# Repetitive Self-Inflicted Craniocerebral Injury in a Patient with Antisocial Personality Disorder

**DOI:** 10.3390/diagnostics14141549

**Published:** 2024-07-18

**Authors:** Andrei Ionut Cucu, Claudia Florida Costea, Sînziana Călina Silișteanu, Laurentiu Andrei Blaj, Ana Cristina Istrate, Raluca Elena Patrascu, Vlad Liviu Hartie, Emilia Patrascanu, Mihaela Dana Turliuc, Serban Turliuc, Anca Sava, Otilia Boişteanu

**Affiliations:** 1Faculty of Medicine and Biological Sciences, University Stefan cel Mare of Suceava, 720229 Suceava, Romania; sinziana.silisteanu@usm.ro; 2Emergency Clinical Hospital Prof. Dr. Nicolae Oblu, 700309 Iasi, Romania; laurentiu-andrei.blaj@d.umfiasi.ro (L.A.B.); istrate.ana-cristina@d.umfiasi.ro (A.C.I.); liviu.vlad.hartie@umfiasi.ro (V.L.H.); turliuc_dana@yahoo.com (M.D.T.); dr.anca.sava.68@gmail.com (A.S.); 3Department of Ophthalmology, University of Medicine and Pharmacy Grigore T. Popa Iasi, 700115 Iasi, Romania; 4Department of Neurosurgery, University of Medicine and Pharmacy Grigore T. Popa Iasi, 700115 Iasi, Romania; 5National Institute for Infectious Diseases Prof. Dr. Matei Bals, 021105 Bucharest, Romania; raluca.jipa1@drd.umfcd.ro; 6Infectious Diseases Department, Faculty of Medicine, University of Medicine and Pharmacy Carol Davila, 050474 Bucharest, Romania; 7Department of Anesthesia and Intensive Care, University of Medicine and Pharmacy Grigore T. Popa Iasi, 700115 Iasi, Romania; patrascanu.emilia@umfiasi.ro (E.P.); otilia.boisteanu@umfiasi.ro (O.B.); 8Regional Institute of Oncology, 700483 Iasi, Romania; 9Department of Psychiatry, Faculty of Medicine, University of Medicine and Pharmacy Grigore T. Popa Iasi, 700115 Iaşi, Romania; serban_turliuc@yahoo.com; 10Department of Morpho-Functional Sciences I, Faculty of Medicine, University of Medicine and Pharmacy Grigore T. Popa Iasi, 700115 Iași, Romania; 11Sf. Spiridon County Clinical Emergency Hospital Iași, 700111 Iași, Romania

**Keywords:** intracranial foreign body, holesurub, nail, self-inflicted craniocerebral injury

## Abstract

Self-inflicted penetrating injuries in patients with mental disorders are a rare phenomenon. The authors report the case of a prisoner who recurrently presented to the emergency department over a period of four years for self-insertion of six metal foreign bodies into the skull. Computed tomography each time revealed the presence of a metal foreign body (screw, nail, metal rod, and wire) passing through the frontal bone into the frontal lobe. In each situation, the foreign body was safely extracted with a favorable outcome. Despite the use of the latest imaging modalities, metal artifacts can limit the assessment of vascular involvement, and special attention must be given to preoperative planning. Surgical extraction of the foreign body can be safely performed when appropriate preoperative planning is carried out to consider all possible complications.

## 1. Introduction

Self-stabbing of the brain is an unusual form of self-harm, and due to the fact that the brain is an organ almost entirely protected by bone, the only regions through which it can be directly accessed are the optic canal and orbital fissures [1]. On the other hand, in contrast with self-stabbing of the brain, a self-inflicted gunshot injury to the brain represents a method of suicide, and only a small portion of those who do this are considered to be psychotic [2]. Although there are numerous neurosurgical reports of foreign intracranial insertion, self-inflicted penetrating intracranial injuries are rare and have been reported in patients with mental disorders, those in custody for serving a prison sentence, or both [3,4,5,6], with some authors reporting that confinement in jail appears to be associated with self-stabbing [7]. The majority of patients with penetrating intracranial nail injuries present without neurological deficits, and death in such cases is rare [8]. Voluntary intracranial nail or screw insertion is a neurosurgical emergency, and surgical extraction can be successfully performed when appropriate preoperative planning takes into account all possible complications. Blind removal of the foreign body is unacceptable due to the risk of secondary brain injuries [9]. We report the case of a 33-year-old man with antisocial personality disorder who recurrently hammered screws into his skull and ingested three other metal foreign bodies over a period of four years. The literature on self-inflicted injury and surgical management of these patients is reviewed.

## 2. Case Report

A 30-year-old man with a mental illness first presented in our department of neurosurgery in July 2020 with a metal foreign body (screw) inserted into the frontal bone, without penetrating the intracranial space (Figure 1A). There was no alteration in the conscious level and no neurological deficit. From the patient’s history, we learned that he had an organic personality disorder of the antisocial type, for which he was receiving treatment from a psychiatrist with Quetiapine 400 mg (200 mg in the morning and 400 mg in the evening), Trihexyphenidyl 2 mg (2 mg in the evening), Valproate 10 mg (10 mg in the morning and 10 mg in the evening), and Tiapride 100 mg (100 mg in the morning and 100 mg in the evening).

The head scanogram and CT scan revealed the embedding of a screw in the frontal bone, immediately above the frontal sinus (Figure 1A). Considering that the metallic foreign body (screw) was only embedded in the frontal bone (Figure 1A), its removal was performed under local anesthesia. The foreign body was safely removed without difficulty and the wound was debrided and closed. At the same time, broad-spectrum antibiotic therapy was initiated for 14 days, and the patient was discharged safely.

Three years later (April 2023), the patient was re-admitted having hammered a 5.6 cm screw through the frontal scar and thinned skull (Figure 1B). GCS was 15 without any neurological deficit or any local or general signs of infection.

At the time of admission, the patient was afebrile and presented a few hours after his screw insertion into the skull. On clinical examination, the exposed end of the screw was observed in the middle frontal region. As in the first presentation, the foreign body was removed, the wound was debrided, and broad-spectrum antibiotic therapy was initiated.

Subsequently, in May 2023 and August 2023, the patient again presented to our department for self-insertion through the same defect in the frontal bone of a metal rod with a length of 4.81 cm (Figure 2A) and a screw with a length of 2.5 cm (Figure 2B), respectively. In both cases, the foreign body was surgically removed, and prophylactic antibiotic therapy was administered, with a good prognosis for the patient.

Two months later, in October 2023, the patient presented again to our service for the insertion of a new metal rod with a length of 3.8 cm through the pre-existing frontal bone defect (Figure 3A–C). Additionally, the patient ingested two metal objects (a broken nail clipper) that passed without further complications, and radiography did not confirm any intra-abdominal residual foreign bodies (Figure 3D,E). Under general anesthesia, the foreign body was extracted (Figure 3C), and broad-spectrum antibiotics were administered, including Ceftriaxone 2 g every 12 h and Clindamycin 600 mg every 8 h for two weeks.

In May 2024, the patient presented again for the insertion of a metal rod with a length of 4.13 cm intracranially through the pre-existing frontal bone defect (Figure 4).

Each time upon admission, the Glasgow Coma Score was 15/15, and a visible end of a metal object was observed in situ in the middle frontal region. Vital signs were normal, and the patient did not present any focal neurological deficits or signs of local or systemic infection. Since the patient was institutionalized, he was detected early and brought to the hospital in a timely manner.

Upon clinical inspection, a different foreign body (screw, nail, metal rod, and wire) was found fixed in the frontal bone. The patient had a history of organic personality disorder (personality disorder with self-mutilation) for which he was receiving treatment, which I mentioned above. Each time, the patient acquired the metal object and in a fit of rage proceeded to insert it through the glabella region into his brain. The patient remembered each instance and reported that he did not intend to die by suicide. Subsequent plain skull X-rays and head CT consistently revealed metal foreign bodies of varying sizes passing through the frontal bone in the midline region.

Informed consent was obtained each time, and under general or local anesthesia, the foreign body was gently and carefully extracted from the intracranial region using surgical forceps. The surgical field was dry, without any bleeding at the operative site. The edge of the cranial bone at the penetration point was cleaned and debrided, and the intraoperative site was irrigated with saline solution and gentamicin. Postoperative head CT scans confirmed the absence of residual foreign bodies. Additionally, the patient received a tetanus vaccine. The patient was prophylactically treated with intravenous antibiotics and anticonvulsants. Each time, he remained neurologically intact postoperatively. Although the patient was offered the option to cover the bone defect with a titanium mesh each time, he did not agree. The patient recovered without incident and was discharged safely with a favorable outcome.

## 3. Material and Methods

To compare our case with similar cases in the literature, we searched the most well-known databases (PubMed, Science Direct, SpringerLink, and Web of Science) using the keywords intracranial and foreign body. After identifying 980 articles, we excluded the gunshot injuries and included only cases of self-inflicted intracranial foreign body in our study, with a total of 25 cases identified. Among these, we selected only cases of self-inflicted intracranial foreign body involving only the frontal region, resulting in the identification of 11 cases that were included in our study (Figure 5).

## 4. Discussions

This article reports a recurrent case of self-flagellation by the repetitive intracranial insertion of six metal foreign bodies by a person with mental illness, occurring over a period of four years.

Self-inflicted injury of the head with a thin metal object (nail, screw, wire, metal rod, etc.) represents a rare idiosyncratic phenomenon [1,10,11,12,13,14,15]. The most common self-inflicted head injuries are found in patients with paranoid schizophrenia, and the injuries are often inflicted to silence inner voices [16], obey auditory hallucinations [17], or attempt suicide [1,12,13]. Various authors have reported that psychotic illness appears to be more frequent in those who stab themselves in the brain compared to those who shoot themselves. For example, in his series of 50 individuals, Weinberger et al. reported that only 2 out of 50 individuals who died by suicide by self-stabbing in the head had a psychiatric illness diagnosis [2]. Similarly, Frierson and Lippmann found that 8.5% of 260 patients who survived a self-inflicted gunshot wound were diagnosed with schizoaffective disorder or schizophrenia [18]. In a multicenter study conducted in England involving 58 self-inflicted gunshot victims, Sutton et al. reported that 4.2% of patients had psychotic disorders [19]. It is noted that psychotic illnesses are more common among patients who stab themselves in the brain or other regions of the body. In this regard, Fukube et al. reported that 17% of 65 self-stabbers had a history of psychiatric illness [20], and Abdullah et al. found a similar percentage of 17% with a diagnosis of psychosis among 23 patients with severe abdominal injuries [21].

In a review that included eight cases of self-inflicted intracranial injuries from the Polish literature, Green et al. reported that all patients had a history of severe psychopathology, and four of these patients hammered nails into their skulls repeatedly, from two to five times [4]. In our case, the patient inserted six metallic objects through the defect in the frontal bone. A similar case was reported by Musa et al. in 1997 of a young man with a severe mental disorder who self-inserted six metallic foreign objects, nails, into the intracranial space [3]. Additionally, the literature reports cases where foreign bodies in the head were accidentally discovered, with patients being unaware of their presence [14]. Some intracranial foreign bodies have been discovered over fifty years later, prompting discussions about infanticide [22].

Although some patients use anatomical regions with weak resistance points such as the bony fissures of the orbit, the cribriform plate of the ethmoid bone, or natural orifices like the mouth or nostrils, according to some authors, hammering through the vertex is the predominant entry site [3].

Interestingly, the majority of patients with psychotic disorders who attempt suicide by inserting a metal nail near the midline do not experience lasting consequences [14,23]. In addition to nails, the specialized literature has reported various objects such as knives, screwdrivers, wrenches, scissors, or spears [9,24,25], but among these, nails and screws are the most commonly used [5] (Table 1). Our patient used six types of metal objects: screws, nails, wire, and various types of metal rods.

Impalements, whether self-inflicted or not, usually result in low-velocity injuries and are not associated with cavitation, shock wave production, or diffuse axonal injuries [35]. In such cases, injuries are typically localized along the trajectory of the foreign body [36].

### 4.1. Imaging

In cases of a penetrating head injury by a foreign body, a noncontrast CT scan is recommended, as studies have shown that this is the primary modality used for initial imaging evaluation [37]. Thus, CT imaging highlights both the position and trajectory of the foreign body, as well as associated brain injuries [38,39]. Although CT scanning is a widely used examination today, it is not without limitations. Specifically, CT imaging of metallic objects has the disadvantage of metal artifacts, which can limit the assessment of vascular involvement [10]. In such cases, Gupta et al. recommend the use of dual-energy computed tomography, which allows for better suppression of streak artifacts [11].

In the case of foreign bodies inserted along the midline of the frontal bone, the greatest risk is injury to the superior sagittal sinus. Other vascular structures that may be injured include the proximal segments of the anterior cerebral arteries (especially the precallosal and supracallosal segments), as well as the callosomarginal artery or its branches. When significant hemorrhages are identified on native CT examination, or when there is suspicion of cerebral artery injury, cerebral angiography is recommended in such situations to rule out acute vascular injuries, traumatic pseudoaneurysm, or fistula occurrence [40]. In these situations, angiography can be useful, as it can describe the anatomy of a vessel and its involvement, or most importantly, it can describe the coexistence of a pseudoaneurysm, the rupture of which can be catastrophic during foreign body extraction [9]. MRI scanning is indicated only when the metallic nature of the foreign body has been fully ruled out.

### 4.2. Treatment

The standard management of penetrating intracranial injury involves a comprehensive assessment of the depth and location of the foreign body, followed by the most appropriate surgical decision for foreign body extraction [22]. Surgical treatment in such cases varies from simple removal of the foreign body and local debridement to craniotomy with extensive cerebral debridement. Several studies in the literature have reported craniotomy as the surgical procedure in 8 out of 13 patients [4,33,34,41,42,43,44]. In some rare cases where foreign bodies are incidentally discovered, conservative management may be considered [45]. However, leaving foreign bodies in situ entails several risks, including headaches [46], infections [47], seizures [48,49,50], and migration of foreign bodies with subsequent neurological damage [51].

In our case, we decided each time to perform surgical extraction of the foreign body. Under general or local anesthesia, we performed debridement of the tissue around the entry site and enlarged the entry hole. Subsequently, we extracted the foreign body, attempting to maintain the trajectory as much as possible to avoid causing additional damage. The foreign body detached easily and was removed without any resistance. Afterward, the trajectory space was abundantly irrigated with normal saline and gentamicin, and the wound was debrided and sutured. To prevent cerebrospinal fluid leaks and infections, some authors recommend surgical exploration and thorough irrigation [10].

The patient received dual antibiotic therapy for 14 days (ciprofloxacin and clindamycin). Some authors recommend administering antibiotics for at least 1–2 weeks [9]. There were no major complications such as meningitis, brain abscess, dural sinus thrombosis, or cerebrospinal fluid leaks.

Managing these patients can sometimes be difficult due to the underlying psychiatric disorder. In a series of eight cases, Green et al. reported one patient’s refusal to have the metal object extracted, and this patient subsequently died due to a brain abscess [4]. Our patient presented a similar challenge, but with continuous communication with the physician and a cooperative attitude, he accepted surgical treatment for foreign body extraction each time.

## 5. Conclusions

In such cases, comprehensive imaging studies and proper preoperative planning are extremely important steps to achieve good outcomes. However, metal artifacts generated by an intracranially implanted metal foreign body from a self-inflicted injury can limit the assessment of possible vascular involvement, and therefore, careful evaluation is essential to prevent secondary brain injuries. Important considerations must be given to psychopathology and the possibility of self-injurious behavior, which is recurrent. However, it is essential to remember that surgical management is secondary, while psychiatric management remains the cornerstone in the long-term care of these patients.

## Figures and Tables

**Figure 1 diagnostics-14-01549-f001:**
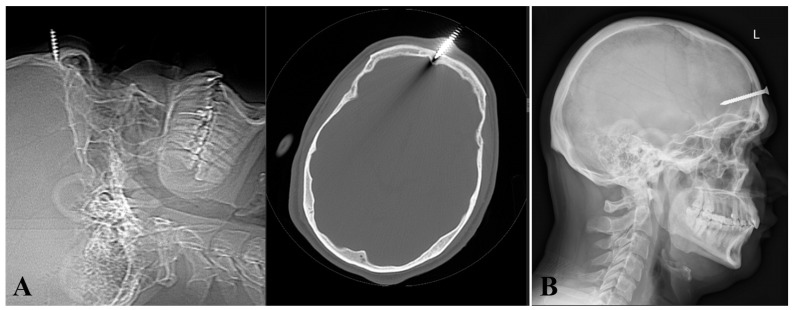
(**A**) Head scanogram and CT reveal the presence of a screw in the frontal bone, without reaching the intracranial space (first injury). (**B**) Skull radiography shows a 5.6 cm screw entering through the frontal bone into the brain (second injury).

**Figure 2 diagnostics-14-01549-f002:**
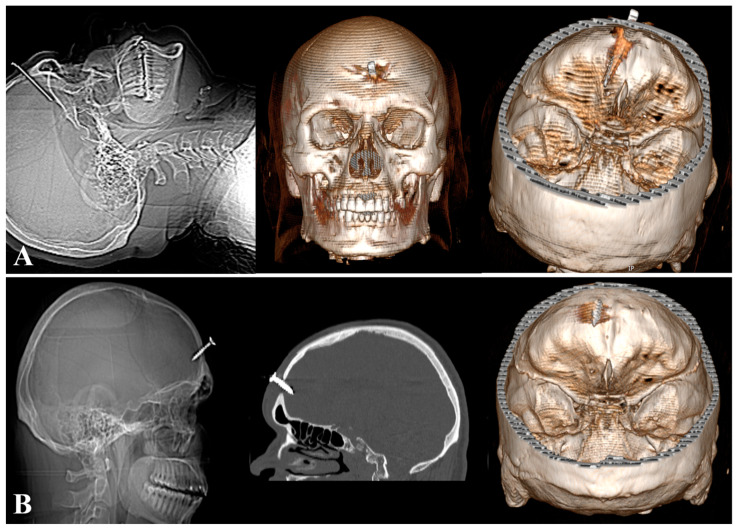
(**A**) Head scanogram and 3D CT showing the intracranial position of the foreign body (metal rod) (third injury). (**B**) Head scanogram, CT scan, and 3D CT showing the position of the foreign body (screw) (fourth injury).

**Figure 3 diagnostics-14-01549-f003:**
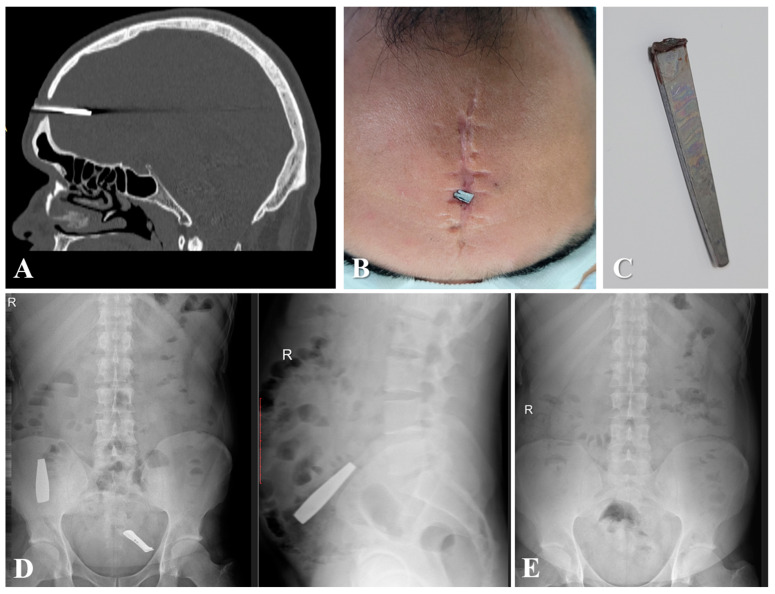
(**A**) Head CT-scan showing the intracranial foreign body (metal rod) (fifth injury). (**B**) Site of insertion of the foreign body and old scar. (**C**) Postoperative image with the extracted foreign body. (**D**) Abdominal radiographs showing two intra-abdominal metal foreign bodies. (**E**) Follow-up abdominal radiograph showing the absence of metal foreign bodies.

**Figure 4 diagnostics-14-01549-f004:**
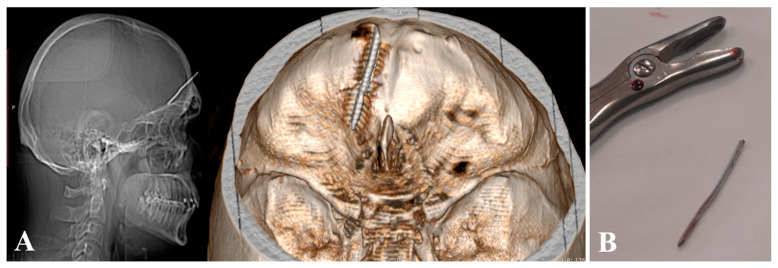
(**A**) Head scanogram and 3D CT showing the metal foreign body (a wire) (fifth injury). (**B**) Postoperative image with the extracted foreign body.

**Figure 5 diagnostics-14-01549-f005:**
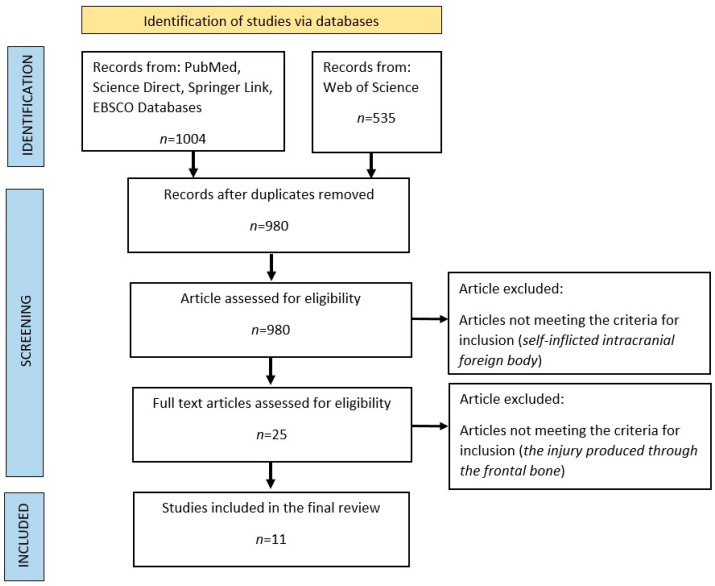
PRISMA flowchart.

**Table 1 diagnostics-14-01549-t001:** Case reports of intracranial self-inflicted intracranial insertion of foreign bodies through the frontal bone, after Large et al. [1].

Author, Year	Age/Gender	Foreign Body	Psychiatric Diagnosos	Medical Outcome
Saint-Martin et al., 2008 [17]	24/M	Knife	Schizophrenia	Death
Gökçek C et al., 2007 [26]	20/M	Nail	No data	Recovered
Karabatsou et al., 2005 [27]	32/M	Wire	Bipolar affective disorder	Recovered
Mitra et al., 2002 [28]	24/M	Knife	Schizophrenia	Recovered
Musa et al., 1997 [3]	28/M	Nail	Personality disorder	Epilepsy
Puri et al., 1994 [29]	30/M	Nail	Psychosis	Recovered
Abe et al., 1986 [30]	25/M	Ice pick	Depression	Recovered
Dempsey and Hoff, 1977 [31]	27/M	Knife	Schizophrenia	Recovered
Fox and Branch, 1971 [32]	50/M	Two nails	Depression	Motor deficit
Azariah, 1970 [33]	19/M	Three wires	Unknown	Death
Reeves, 1965 [34]	Adult/F	Nail	Psychosis	Death

## Data Availability

All data are reported in the text.

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
