# Peer review of "Repetitive Self-Inflicted Craniocerebral Injury in a Patient with Antisocial Personality Disorder"

_diagnostics, 2024, doi:10.3390/diagnostics14141549_

Round 1
Reviewer 1 Report
Comments and Suggestions for Authors
The authors reported a case with psychotic disorder, who repeated self-stabbing of the brain. Self-stabbing wound of the brain is rare, but enough reports are piled in the literature, but repeated injury seems unusual. As they mentioned, treatment of the underlying psychotic disorder is primarily important.
They say “neurological deficits and infections due to these intracranial foreign bodies 171 are almost never reported.” But there may be publication bias about this point.
Based on their experience, the burr hole or gap between the bone and flap should have covered with metal or some materials to raise the hurdle of re-stubbing, though it may increase the risk of infection.
Author Response
|
Response to Reviewer 1 Comments
|
||
|
|
|
|
|
Thank you very much for taking the time to review this manuscript. Please find the detailed responses below and the corresponding revisions/corrections highlighted/in track changes in the re-submitted files.
|
||
|
Comments 1: The authors reported a case with psychotic disorder, who repeated self-stabbing of the brain. Self-stabbing wound of the brain is rare, but enough reports are piled in the literature, but repeated injury seems unusual. As they mentioned, treatment of the underlying psychotic disorder is primarily important. |
||
|
Response 1: Thank you for pointing this out. We agree with this comment. Although a series of similar cases are reported in the literature, the particularity of our case is that the patient inserted nails into the intracranial space multiple times within a relatively short period. Indeed, one of the important conclusions of the article is the proper and responsible management of the underlying psychiatric illness to prevent such self-stabbing injuries of the brain
|
||
|
Comments 2: They say “neurological deficits and infections due to these intracranial foreign bodies 171 are almost never reported.” But there may be publication bias about this point. Response 2: In the original source, the phrase referred to small objects like nails. But we removed the statement from our text, as it could indeed represent a type of bias. Thank you for your attention!
Comments 3 Based on their experience, the burr hole or gap between the bone and flap should have covered with metal or some materials to raise the hurdle of re-stubbing, though it may increase the risk of infection. Response 3: Indeed, each time, and especially when the opening became larger than the diameter of a nail, the patient was offered the placement of a titanium plate (titanium mesh) at the site of the hole, but each time he refused this intervention. We have also added these new aspects to the manuscript text to make it more informative. Thank you for the addition.
|
||
Thank you for the valuable comments that further enhance our article and for the important remarks.
Reviewer 2 Report
Comments and Suggestions for Authors
The paper reports about self-inflicted lesion from foreign body. The Tepic is unusual and, despite this occurrence is very rare in the clinical practice, contains significative cultural information useful for the general audience. The paper is well written and understandable. I have nike to report significant criticisms to this paper. I only suggest authors to insert a short paragraph with method for literature review and a prisma diagram for self-inflicted intracranial injuries both metal items or gunshot.
Comments on the Quality of English LanguageEnglish language is readable and easily understandable and only need minor adjustment
Author Response
|
Response to Reviewer 2 Comments
|
||
|
|
|
|
|
Thank you very much for taking the time to review this manuscript. Please find the detailed responses below and the corresponding revisions/corrections highlighted/in track changes in the re-submitted files.
|
||
|
Comments 1: The paper reports about self-inflicted lesion from foreign body. The Tepic is unusual and, despite this occurrence is very rare in the clinical practice, contains significative cultural information useful for the general audience. The paper is well written and understandable |
||
|
Response 1: The authors would like to take this opportunity to thank the reviewer for the words of appreciation. Indeed, it was challenging to maintain a logical flow of the case since the patient repeatedly inserted a total of six metal objects into his skull over a period of four years. We are grateful for the words of appreciation.
|
||
|
Comments 2: I have nike to report significant criticisms to this paper. I only suggest authors to insert a short paragraph with method for literature review and a prisma diagram for self-inflicted intracranial injuries both metal items or gunshot. Response 2: We also added a section on materials and methods in the text, as well as a PRISMA flowchart. We note that our study included only self-inflicted intracranial foreign bodies (metal items) and not objects introduced by gunshot. Thank you once again for your valuable contribution. |
||
Thank you for the valuable comments that further enhance our article and for the important remarks.